# Effectiveness of Geriatric Assessment-Driven Interventions on Survival and Functional and Nutritional Status in Older Patients with Head and Neck Cancer: A Randomized Controlled Trial (EGeSOR)

**DOI:** 10.3390/cancers14133290

**Published:** 2022-07-05

**Authors:** Elena Paillaud, Lydia Brugel, Chloe Bertolus, Melany Baron, Emilie Bequignon, Philippe Caillet, Thomas Schouman, Jean Lacau Saint Guily, Sophie Périé, Eric Bouvard, Marie Laurent, Didier Salvan, Laurence Chaumette, Laure de Decker, Benoit Piot, Beatrix Barry, Agathe Raynaud-Simon, Elisabeth Sauvaget, Christine Bach, Antoine Bizard, Abderrahmane Bounar, Aurelien Minard, Bechara Aziz, Eric Chevalier, Dominique Chevalier, Cedric Gaxatte, Olivier Malard, Evelyne Liuu, Sandrine Lacour, Laetitia Gregoire, Charlotte Lafont, Florence Canouï-Poitrine

**Affiliations:** 1IMRB, Institut National de la Santé et de la Recherche Médicale (INSERM), Université Paris-Est Créteil, F-94010 Créteil, France; philippe.caillet@aphp.fr (P.C.); marie.laurent@aphp.fr (M.L.); charlotte.lafont@aphp.fr (C.L.); florence.canoui-poitrine@aphp.fr (F.C.-P.); 2Département de Gériatrie, Paris Cancer Institute CARPEM, Hôpital Européen Georges Pompidou, Assistance-Publique Hôpitaux de Paris (AP-HP), F-75015 Paris, France; 3Service d’ORL et Chirurgie Cervico-Faciale, Centre Hospitalier Intercommunal de Créteil, F-94010 Créteil, France; lydia.brugel@chicreteil.fr (L.B.); emilie.bequignon@chicreteil.fr (E.B.); 4Service de Chirurgie Maxillo-Faciale, Sorbonne Université, Hôpital Pitié Salpétrière, Assistance-Publique Hôpitaux de Paris (AP-HP), F-75013 Paris, France; chloe.bertolus@aphp.fr (C.B.); thomas.schouman@aphp.fr (T.S.); 5Service de Soins de Suites et de Réadaptation Gériatrique, Sorbonne Université, Hôpitaux Universitaires Pitié Salpêtrière-Charles Foix, Assistance-Publique Hôpitaux de Paris (AP-HP), F-94205 Ivry sur Seine, France; melany.baron@aphp.fr; 6Département d’ORL, Sorbonne Université, Hôpital Tenon, Assistance-Publique Hôpitaux de Paris (AP-HP), F-75020 Paris, France; prlacaustguily@gmail.com (J.L.S.G.); prsophieperie@gmail.com (S.P.); 7Département d’ORL, J Lacau St Guily Exerce à l’Hôpital-Fondation Rothschild, S Périé Exerce à la Clinique Hartmann, F-92200 Neuilly Sur Seine, France; 8Service de Gériatrie, Hôpital Tenon, Assistance-Publique Hôpitaux de Paris (AP-HP), F-75020 Paris, France; eric.bouvard@aphp.fr; 9Département de Gériatrie, Hôpital Henri-Mondor, Assistance-Publique Hôpitaux de Paris (AP-HP), F-94010 Créteil, France; 10Service ORL et Cervico-Facial, Centre Hospitalier Sud Francilien, F-91100 Corbeil-Essonnes, France; didier.salvan@chsf.fr; 11Service de Court Sejour Gériatrique, Centre Hospitalier Sud Francilien, F-91100 Corbeil-Essonnes, France; laurence.chaumette@chsf.fr; 12Service de Gériatrie, Centre Hospitalier Universitaire de Nantes, F-44093 Nantes, France; laure.dedecker@chu-nantes.fr; 13Service de Chirurgie Maxillo-Faciale et Stomatologie, Centre Hospitalier Universitaire de Nantes, F-44093 Nantes, France; benoit.piot@chu-nantes.fr; 14Service ORL et Chirurgie Cervico-Faciale, Université de Paris, Hôpital Bichat, Assistance-Publique Hôpitaux de Paris (AP-HP), F-75018 Paris, France; beatrix.barry@aphp.fr; 15Service de Gériatrie, Université Paris Cité, Hôpital Bichat, Assistance-Publique Hôpitaux de Paris (AP-HP), F-75018 Paris, France; agathe.raynaud-simon@aphp.fr; 16Service ORL et Chirurgie Cervico-Faciale, Groupe Hospitalier Paris-Saint Joseph, F-75014 Paris, France; esauvaget@hpsj.fr; 17Service d’ORL, Hôpital Foch, F-92150 Suresnes, France; c.bach@hopital-foch.org; 18Unité de Gériatrie Aigue, Hôpital Foch, F-92150 Suresnes, France; a.bizard@hopital-foch.com; 19Unité de Gériatrie Aigue, Centre Hospitalier Intercommunal Villeneuve-Saint-Georges, F-94190 Villeneuve-Saint-Georges, France; abderrahmane.bounar@chiv.fr; 20Service de Gériatrie, Hôpital Léopold Bellan, F-75014 Paris, France; aurelien.minard@fondationbellan.org; 21Service ORL et Chirurgie Cervico-Faciale, Centre Hospitalier Intercommunal Villeneuve-Saint-Georges, F-94190 Villeneuve-Saint-Georges, France; bechara.aziz@chiv.fr; 22Service ORL et Chirurgie Cervico-Faciale, Groupement Hospitalier Intercommunal Le Raincy-Montfermeil, F-93370 Montfermeil, France; eric.chevalier@ght-gpne.fr; 23Service ORL et Chirurgie Cervico-Faciale, Centre Hospitalier Universitaire de Lille, Hôpital Huriez, F-59000 Lille, France; dominique.chevalier@chru-lille.fr; 24Service de Médecine Gériatrique, Centre Hospitalier Universitaire de Lille, F-59000 Lille, France; cedric.gaxatte@chru-lille.fr; 25Service d’ORL et de Chirurgie Cervico-Faciale, Centre Hospitalier Universitaire de Nantes, F-44093 Nantes, France; olivier.malard@chu-nantes.fr; 26Service de Gériatrie, Centre Hospitalier Universitaire de Poitiers, F-86021 Poitiers, France; evelyne.liuu@chu-poitiers.fr; 27Centre de Recherche Clinique, Centre Hospitalier Intercommunal de Créteil, F-94010 Créteil, France; sandrine.lacour@chicreteil.fr; 28Unité de Recherche Clinique (URC-Mondor), Hôpital Henri-Mondor, AP-HP, F-94010 Créteil, France; laetitia.gregoire@aphp.fr; 29Service de Santé Publique, Hôpital Henri-Mondor, Assistance-Publique Hôpitaux de Paris (AP-HP), F-94010 Créteil, France

**Keywords:** geriatric assessment, head and neck cancer, elderly, survival, nutrition, function

## Abstract

**Simple Summary:**

Due to population ageing, there is an increasing number of older patients with head and neck cancers (HNC). Management of HNCs is complex. This population may be frailer than other patients with solid cancer. The Geriatric Assessment (GA) is a multidimensional diagnostic and therapeutic tool focused on frailty to propose a coordinated treatment plan and long-term follow-up. Several trials assessed the efficacy of GA-driven interventions on diverse outcomes but no recent randomized controlled trial demonstrated the impact on mortality, functional, or nutritional status as a primary outcome in this particular population. This trial highlighted several difficulties in implementation of geriatric interventions and suggested that the assessment of other models as co-management with oncologists and/or experienced practice nurses could be useful in clinical routine practice.

**Abstract:**

This study assesses the efficacy of Geriatric Assessment (GA)-driven interventions and follow-up on six-month mortality, functional, and nutritional status in older patients with head and neck cancer (HNC). HNC patients aged 65 years or over were included between November 2013 and September 2018 by 15 Ear, Nose, and Throat (ENT) and maxillofacial surgery departments at 13 centers in France. The study was of an open-label, multicenter, randomized, controlled, and parallel-group design, with independent outcome assessments. The patients were randomized 1:1 to benefit from GA-driven interventions and follow-up versus standard of care. The interventions consisted in a pre-therapeutic GA, a standardized geriatric intervention, and follow-up, tailored to the cancer-treatment plan for 24 months. The primary outcome was a composite criterion including six-month mortality, functional impairment (fall in the Activities of Daily Living (ADL) score ≥2), and weight loss ≥10%. Among the patients included (*n* = 499), 475 were randomized to the experimental (*n* = 238) or control arm (*n* = 237). The median age was 75.3 years [70.4–81.9]; 69.5% were men, and the principal tumor site was oral cavity (43.9%). There were no statistically significant differences regarding the primary endpoint (*n* = 98 events; 41.0% in the experimental arm versus 90 (38.0%); *p* = 0.53), or for each criterion (i.e., death (31 (13%) versus 27 (11.4%); *p* = 0.48), weight loss of ≥10% (69 (29%) versus 65 (27.4%); *p* = 0.73) and fall in ADL score ≥2 (9 (3.8%) versus 13 (5.5%); *p* = 0.35)). In older patients with HNC, GA-driven interventions and follow-up failed to improve six-month overall survival, functional, and nutritional status.

## 1. Introduction

With aging of the population, increasing numbers of older patients are suffering from head and neck cancers (HNC) [1]. About 30% of HNCs are diagnosed in patients aged 70 years or over and 10% over 80 years of age [1,2].

The management of HNC in older patients is complex because of the high toxicity of loco-regional treatments (notably affecting swallowing and communication after surgery) and no treatment standards because of a lack of dedicated studies and the heterogeneity of older people [3,4,5]. Comorbidities, impaired nutritional, mood, cognition and functional status, and a poor social environment are associated with treatment complications and decreased overall survival [6,7,8,9,10]. Moreover, HNC patients may be frailer than other patients with solid tumors because of their major functional impacts [11]. The assessment of frailty by geriatric assessment (GA) is recommended by learned societies to evaluate the capacity of older patients to undergo cancer treatment [12,13]. The Comprehensive Geriatric Assessment (CGA) was developed by geriatricians as a ‘multidimensional interdisciplinary diagnostic process focused on determining a frail older person’s medical, psychological, and functional ability in order to develop a coordinated and integrated plan for treatment and long-term follow-up’ [14]. The CGA offers both a diagnostic and therapeutic tool. In a non-cancer setting, a CGA increases the probability of being alive at home [15]. In older patients with cancer, several recent randomized trials showed the benefit of GA-driven intervention on chemotherapy and toxics effects as primary outcomes but did not find any impact of GA-driven intervention on mortality as a secondary outcome [16,17,18]. To our knowledge, no randomized controlled trial has demonstrated any beneficial effects of GA-driven interventions and geriatric follow-up on mortality or daily living activities (functional status) or nutritional status as a primary outcome in older patients with cancer.

We hypothesized that a GA-driven intervention and follow-up in older patients with Head and Neck Squamous Cell Carcinoma (HNSCC) could improve overall survival and functional and nutritional status. Our main objective was to evaluate the benefits of GA-driven interventions and follow-up on six-month overall survival and functional and nutritional status in older patients with HNSCC.

## 2. Materials and Methods

### 2.1. Experimental Design

This was an open-label, multicenter, randomized, controlled trial in two parallel groups and independent outcome assessors in older patients with HNC benefiting from GA-driven interventions and follow-up (experimental arm) versus standard of care (control arm).

### 2.2. Setting

The trial was conducted in 13 centers in France; ten in teaching hospitals and three in non-teaching hospitals, and 15 clinical departments: 11 Ear, Nose, and Throat (ENT) and four oral and maxillofacial surgery.

### 2.3. Participants

The eligibility criteria for patients were age ≥70 years, and a macroscopic diagnosis of HNC (oral, oropharyngeal, hypopharyngeal, or laryngeal locations) awaiting histological confirmation. The exclusion criteria were inmate in a correctional facility; subject to legal guardianship; psychological, familial, social, or geographic conditions that might interfere with conduct of the study; a personal history of head and neck cancer and a rare tumor site (sinonasal or salivary gland). All enrolled patients gave their written informed consent. The protocol was approved by the appropriate Ethics Committee (CPP Ile-de-France I, Paris, France, approval on 20 April 2013; no. 13213). The trial is registered on ClinicalTrials.gov [NCT02025062].

Two important changes were made after the trial had started: firstly, we allowed the inclusion of patients between the ages of 65 and 69 years as accrual was slower than anticipated and these adults with HNC could have a frailty profile [11], and secondly, we permitted the inclusion of patients with a previous history of HNC if it had been treated more than five years previously, as the current episode could be considered as a new cancer [19,20].

### 2.4. Intervention

In the experimental arm, patients benefited from a GA-driven intervention and follow-up, while in the control arm, patients received standard of care. The GA-driven intervention and follow-up were assured by a senior geriatrician with help from a nurse if necessary, and had four components as previously described in detail [21].

Briefly:

1—Pre-treatment geriatric assessment: This was performed by a geriatrician with expertise in oncology before the multidisciplinary meeting during which the cancer treatment plan would be established. When this schedule was not possible for organizational reasons, the geriatric assessment was planned after the multidisciplinary meeting but before the initiation of treatment;

2—Participation of the geriatrician in determining the cancer treatment plan and in the multidisciplinary meeting;

3—A GA-driven intervention proposed by the geriatrician. The intervention program had four components: optimization of the management of problems detected in GA-related health domains and regarding five comorbidities (chronic atrial fibrillation, chronic heart failure, diabetes, coronary artery disease, and hypertension), a medication review, patient education on the self-management of comorbidities, and information on cancer treatments. Corrective measures were implemented as required. Interventions were suggested as recommendations by the geriatrician or directly prescribed to the team in charge of the patient;

4—The standardized geriatric follow-up involved a brief assessment of nutrition, mood, pain, functional status, the five comorbidities listed above, self-perceived health status, medication use, and the implementation of geriatric interventions if necessary. The follow-up schedule consisted in seven medical follow-up visits by geriatricians, the timing of which was adapted to the cancer treatment plan (Supplemental). When it was not possible to organize a face-to-face follow-up visit, the geriatrician completed this over the telephone.

The geriatric domains assessed by the geriatrician at the beginning and during the patient’s follow-up as well as the implemented interventions were presented in Table 2 of the article published on the study protocol in 2014 [21].

### 2.5. Outcomes

The main outcome was a composite criterion, assessed at six months after randomization, and included death, a decrease by two points or more in the Activities of Daily Living (ADL) score from baseline, and weight loss of 10% or more from baseline. Beyond death, loss of independency and weight are major prognostic factors in this population. The ADL and weight measurements were standardized using a measurement guide and a dedicated balance, similar in every center (Seca 813 Robusta, designed in Hamburg (Germany) and made in China). The outcome was adjudicated by an independent clinical research assistant.

### 2.6. Collected Data

Clinical characteristics (gender, weight loss, ADL score, performance status, G8 score based on 8 items: age, appetite change, weight loss, mobility, body mass index, neuropsychological problems, self-rated health, medication), comorbidities, smoking status, and alcohol consumption were collected at baseline. Oncologic characteristics included localization of cancer, TNM stage, and treatment plan (treatment decision based on the multidisciplinary meeting and treatment received). In the experimental arm, characteristics based on the geriatric assessment were collected such as functional status (IADL score), mobility (falls, timed get-up-and-go test, outside walking with help, one-leg standing test), asthenia, comorbidities (total CIRS-G score), nutritional status (MNA score), cognition (MMSE score), and GA-driven interventions.

### 2.7. Sample Size

We hypothesized that the intervention would result in an absolute decrease of at least 10% in the principal endpoint, and we assumed that 30% of controls would achieve the primary endpoint. With a 5% two-sided alpha risk and 80% power, 640 patients would be required for the study (320 in each arm). We assumed that 10% of patients would be lost to follow-up before study completion or that their data would not available regarding the primary endpoint. We therefore planned to include a total of 704 patients.

The first patient was included in November 2013. Due to a slow-down in recruitment and the end of the grant period, the study coordinators convened an interim and independent data monitoring committee (IDMC) meeting in September 2018, at which time, 499 patients had been randomized. Unplanned futility analyses revealed that even if the full number of patients required had been obtained, our results could not have shown a difference in the main outcome, leading to an early termination of the trial.

### 2.8. Randomization

Randomization was centralized using an online system (RandoWeb, Paris, France) [22]. This system applied a minimization program to balance the two arms with respect to center, age (< or ≥80 years), T stage (< or ≥T2), N stage (< or ≥N2), and tumor site (oral, oropharyngeal, hypopharyngeal, or laryngeal). Because simple minimization within centers could, in theory, lead to the alternation of treatment allocations, the algorithm also incorporates 30% of random allocation, thus helping to ensure concealment. The allocation ratio was 1:1.

### 2.9. Statistical Analysis

Baseline demographics, comorbidities, risk factors, and HNSCC characteristics in the two randomized arms were described, as were geriatric characteristics and geriatric interventions in the experimental arm. Quantitative variables were described as means (±1 standard deviation (SD)) or medians (25–75th percentiles) and qualitative variables as numbers (%).

An efficacy analysis on the primary outcome was analyzed using the intent-to-treat principle, with all patients kept in the arm to which they had been assigned by the randomization system and multiple imputation for patients with missing data for the main outcome (i.e., ADL and weight loss changes, not for death criterion). Secondary analyses were performed on complete cases using the per-protocol principle. Logistic regression was used for efficacy analyses under the intention-to-treat and per-protocol principles. Crude Odds ratio (OR) were reported with their 95% confidence intervals (CI). Pearson’s chi-square test was used in a secondary analysis of complete cases. Moreover, each component of the main endpoint was analyzed using a similar methodology (logistic regression and Pearson’s chi-square test), except for death to which survival analysis was applied (log rank test). ADL and weight loss changes were also analyzed continuously and multiple imputations were made for patients with missing data for these variables, before the application of linear regression. 

In the experimental arm, geriatric interventions were generally considered as initiated when at least one of them was prescribed; otherwise they were considered to be suggested, or unnecessary when they were neither prescribed nor suggested.

Predefined subgroup analyses were conducted according to age group (< or ≥80 years), tumor site (oral, oropharyngeal, laryngeal), and metastasis status (M0, M1/Mx).

Statistical significance was considered with *p* < 0.05, and all tests were two-tailed. Statistical analyses were performed with Stata software (version 15.0, StataCorp, College Station, TX, USA), and data were reported according to CONSORT guidelines [23].

## 3. Results

### 3.1. Description of the Population

A total of 499 patients were included in the study, 475 of whom were randomized to benefit from a GA-driven intervention with follow-up (*n* = 238) or not (*n* = 237) (Figure 1). Among these, 176 patients (74%) and 11 (5%) patients did not complete the geriatric and standard follow-up as it was planned, respectively. Patients mostly received the CGA after the multidisciplinary meeting (69%) but 2.5 days in median [0.5–7] right after.

At baseline, the median age was 75.2 years [70.3–82.2] in the experimental arm and 75.6 years [70.6–81.7] in the control arm; 32.8% and 28.3% of patients were women, respectively. Over 60% of patients had a performance status of 0 and less than 15% of patients had an ADL score ≤5.5 in both arms, 59.6% and 64.5% of patients in the experimental and control arms had a G8 score ≤14 (Table 1), respectively. Among patients in the experimental arm, the cancer sites were oral (43.5%), laryngeal (25.0%), oropharyngeal (22.9%), and hypopharyngeal (8.6%), and 20.3% of patients had metastases. In the control arm, the cancer sites were oral (44.4%), laryngeal (24.8%), oropharyngeal (21.7%), and hypopharyngeal (9.1%) and 21% of patients had metastases. Alcohol consumption concerned 38.3% of patients in the experimental arm and 37.5% in the control arm, and included the consumption of wine (60.0% and 47.6% respectively in each arm), strong alcohol (10.6% and 16.7%), and beer (5.9% and 17.9%) (Table 1).

The scheduled cancer treatment (and the actual received treatment) was surgery in 55.2% (92 %) of patients in the experimental arm and 50.8% (98 %) of patients in the control arm, chemotherapy-radiotherapy in 15.2% (74 %) and 14.4% (82 %) of patients respectively, and radiotherapy alone in 12.6% (93 %) and 14.4% (91 %), respectively. Geriatric parameters assessed by the geriatrician during the GA in the experimental arm and the GA-driven interventions are presented in Table 2. In terms of mobility, 11.7% of these patients reported falls during the past six months and 12.4% had an altered timed-get-up-and-go test (>20 s). Asthenia affected 52.8% of patients. As for cognition, 18.2% of patients had an altered MMSE score (<24) and 24% of patients felt depressed. GA-driven intervention were suggested or prescribed by geriatrician for 63% of patients (Table 2).

### 3.2. Efficacy Analysis for the Primary Outcome

There were no statistically significant differences regarding the primary outcome (41.0% versus 38.0%, *p* = 0.53) (Table 3). There was no difference between the two arms when considering each criterion of the primary outcome (Table 3 and Appendix A), nor also when weight loss and ADL were considered in a continuous manner (*p* = 0.88 and *p* = 0.39, respectively). The results were similar for analyses on complete cases, both globally and for each criterion. Under per-protocol analysis, 62 patients were included in the experimental arm and 226 patients in the control arm. No statistically significant differences concerning the primary outcome were observed, either globally or for each criterion.

### 3.3. Subgroup Analyses

There were no statistically significant differences considering GA-driven interventions in the population aged <80 years and the population aged ≥80 years. The results were similar with respect to cancer site and metastases (Figure 2).

## 4. Discussion

The EGeSOR study did not allow to conclude on the impact of GA-driven interventions and follow-up added to standard of care for HNC in older patients on the six-month mortality and functional and nutritional status for several reasons; however, this trial did not demonstrate the effectiveness of GA with or without driven interventions on this outcome.

Five comparable randomized trials concerning other tumor sites have been performed. Recently, Daneng Li et al. found a positive effect of GA-driven interventions in the reduction of chemotherapy toxic effect but did not demonstrate any benefit on survival probabilities after 6 and 12 months in patients with solid tumors in the GAIN trial [18]. Similarly, in a population of patients with colorectal cancer, Lund et al. showed the benefit of GA-based interventions on the completion of chemotherapy but not on mortality [16]. In a cluster-randomized trial, Mohile SG et al. showed a reduction of grade 3–5 chemotherapy toxicities with a GA summary with management recommendations to oncologists in older patients with advanced cancer but no impact on survival [17]. None of these trials considered mortality, functional and nutritional status as a primary outcome. In a subgroup analysis of 99 patients with various tumor sites, Rao AV et al. failed to demonstrate the impact of GA-driven interventions and follow-up on one-year mortality [24]. Similarly, in 122 frail patients operated for colorectal cancer, Ommundsen N et al. did not find any impact of pre-operative GA-driven interventions on the risk of postoperative complications or on secondary outcomes such as repeat surgery, readmission, or mortality [25].

The principal explanation for these results may be the high proportion (74%) of patients who did not benefit from the complete intervention as planned even though all the patients in the intervention group received a GA. This was due to logistical and organizational problems: in four centers, a geriatrician was not present on-site and in four others, the geriatric medicine team lacked the time and personnel to achieve the planned follow-up visits. The lack of geriatricians with expertise in oncology in both France and other countries has been mentioned elsewhere [26,27]. Alternative models (which do not require input from geriatricians) are currently being explored. In another randomized pilot study, the use of an algorithm to guide GA management recommendations to oncologists led to a low level of implementation [26]. Elsewhere, during a randomized controlled trial in older ambulatory post-surgical cancer patients, McCorkle R et al. demonstrated that a standardized home care intervention implemented by advanced practice nurses increased survival [28]. Three other large multicenter randomized trials are currently under way to evaluate the impact of GA-driven interventions implemented by nurse practitioners on chemotoxicity [29] or overall survival and health quality of life [30,31]. The degree of implementation of intervention, i.e., the actual realization of the proposed or recommended intervention, is rarely reported in the studies, but in light of the difficulties encountered, reporting and documenting the effective implementation of interventions is relevant, as it was reported by Daneng Li et al. in the GAIN trial where 76.8 % of interventions were implemented in the experimental arm [18].

The second explanation arises from the role of geriatrician as an advisor. Indeed, a third of GA interventions were only suggested by the geriatrician but we cannot be sure that they were effectively implemented. This was consistent with previous research in both cancer and non-cancer settings where the input of geriatric interventions remained uncertain in the geriatric team model [25,32,33,34]. An alternative model might be co-management, a promising aspect of this model being that geriatric teams would be directly involved in and have direct control over relevant medical issues [35], mainly regarding the treatment of ENT tumors which have a considerable impact on communication and mood. In a before-after cohort study, Shahrokni A et al. found that geriatric co-management with a surgical team was associated with a significant reduction in 90-day postoperative mortality among older patients with cancer [36].

The final explanation may be related to the choice of study population. We chose a large population with few exclusion criteria so that it would be transposable to a real-life setting.

Indeed, we did not include frailty screening in our inclusion criteria and fixed a relatively young cut-off of age as HNC patients are known to be frailer than their peers with other solid malignancies even after adjustment for comorbidities, mainly because their lifestyle is less healthy [11]. However, this led us to include a large proportion of more robust patients, insofar as 38% of our study population had a G8 score >14.

Moreover, the percentages of women, non-smokers and non-alcoholic patients in our study were higher than those anticipated from the ENT cancer registers. This could be explained by an increase in the number of tumors linked to papillomavirus in older patients, which was not classified in our study. It would be interesting to analyze this risk factor because it has a direct impact on survival and treatment. Tumors linked to papillomavirus generally have a better prognosis than those linked to classic risk factors.

The limitations of our study concerned the lack of complete implementation of the intervention, and the fact that we did not attain our planned sample size. Moreover, the choice of endpoints such as mortality, functional limitation and nutritional status could be questionable. It is possible that GA interventions in the context of managing HNC have minor effects on such endpoints, but may improve feasibility, tolerance of anticancer treatment and health-related quality of life. A large multicenter randomized phase III trial is also in progress to evaluate GA-driven interventions and follow-up on the quality of life of patients aged 70+ years with solid tumors, lymphoma or myeloma who are referred for chemotherapy [31]. In addition, due to the heterogeneity and diversity of treatments offered in the study population (i.e., surgery, radiotherapy, chemotherapy), the outcome may probably have been adapted according to the type of treatment. Moreover, it would also have been relevant to collect the information on the initial ENT physician proposal before CGA.

Our study nevertheless had several strengths. Firstly, it was the largest multicenter randomized controlled trial to have assessed the efficacy of GA-driven interventions and follow-up in older patients with cancer on a clinical primary outcome. Secondly, the use of minimization in the randomization method enabled the limitation of selection and confusion bias. Thirdly, the GA-driven intervention was standardized and based on validated international measurement tools and the most recent clinical guidelines.

## 5. Conclusions

We were not able to demonstrate a direct clinical benefit of GA-driven interventions on six-month mortality and functional and nutritional status in older patients with HNC. This implied failure of the intervention model we had chosen. The assessment of other models for GA-driven implementation and follow-up, such as co-management, a shared-care model with oncologists and/or experienced practice nurses could be useful in this setting.

## Figures and Tables

**Figure 1 cancers-14-03290-f001:**
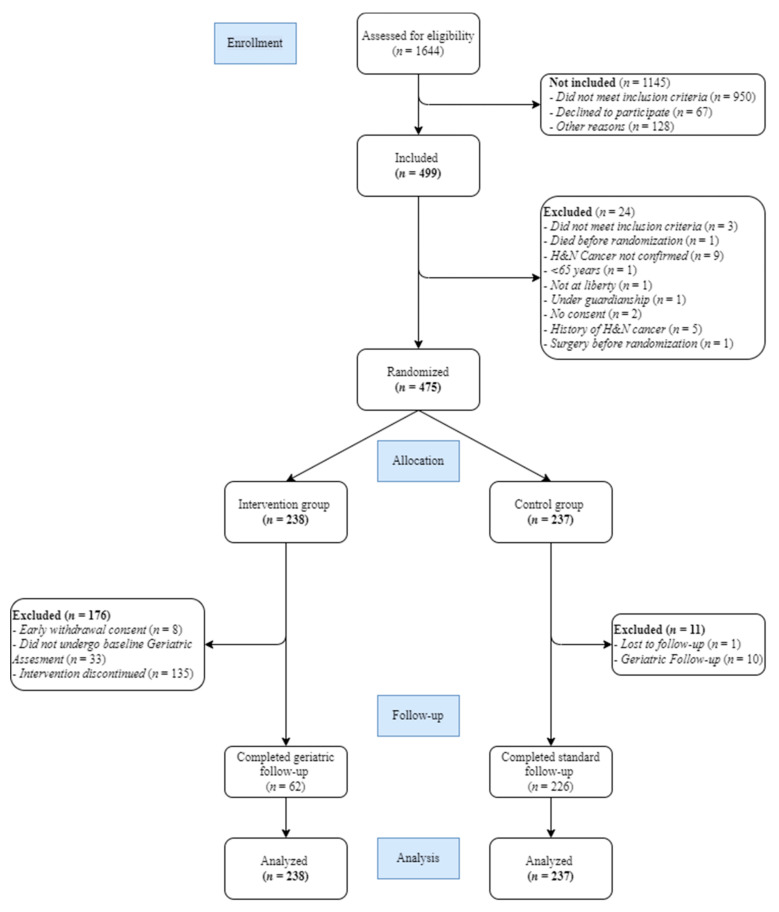
Flowchart for the EGeSOR trial.

**Figure 2 cancers-14-03290-f002:**
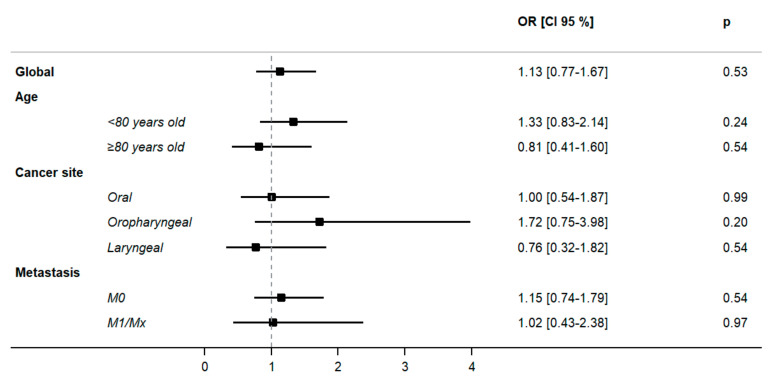
Impact of GA-driven interventions and follow-up on 6-month overall survival and the functional and nutritional status of older patients with HNSCC according to age, cancer site and metastasis status. Logistic regression models were applied. Abbreviations: OR, odds ratio; CI, confidence interval.

**Table 1 cancers-14-03290-t001:** Baseline characteristics of participants.

		Experimental Arm *n* = 238	Control Arm *n* = 237
Characteristics, *n*(%) otherwise indicated			
Age in years, median [Q1–Q3]		75.2 [70.3–82.2]	75.6 [70.6–81.7]
Age groups	[65–70]	57 (23.9)	51 (21.5)
	[70–75]	59 (24.8)	64 (27.0)
	[75–80]	44 (18.5)	48 (20.2)
	[80–85]	42 (17.6)	40 (16.9)
	[85–90]	21 (8.8)	24 (10.1)
	≥90	15 (6.3)	10 (4.2)
Female gender		78 (32.8)	67 (28.3)
Living alone at home (*n* = 227/232)		80 (35.2)	79 (34.1)
Clinical characteristics			
Weight loss ^†^ (*n* = 57/63)		28 (49.1)	27 (42.9)
BMI groups	Malnutrition	39 (16.4)	33 (13.9)
	Normal weight	90 (37.8)	105 (44.3)
	Overweight	75 (31.5)	71 (30.0)
	Obese	34 (14.3)	28 (11.8)
ADL score ≤5.5 (*n* = 233/236)		28 (12.0)	24 (10.2)
Performance status (*n* = 233/236)	0	140 (60.1)	149 (63.1)
	1	71 (30.5)	69 (29.3)
	2	13 (5.6)	9 (3.8)
	3, 4	9 (3.8)	9 (3.8)
G8 score * ≤14 (*n* = 183/186)		109 (59.6)	120 (64.5)
Comorbidities	Hypertension (*n* = 230/236)	134 (58.3)	146 (61.9)
	COPD (*n* = 116/128)	35 (30.2)	41 (32.0)
Cardiac arrhythmia (atrial fibrillation) (*n* = 116/128)		29 (25.0)	33 (25.8)
	Diabetes (*n* = 229/236)	47 (20.5)	40 (17.0)
	PAOD (*n* = 116/128)	22 (19.0)	24 (18.8)
	Heart failure (*n* = 229/236)	28 (12.2)	13 (5.5)
	Stroke (*n* = 116/128)	11 (9.5)	16 (12.5)
	Renal failure (*n* = 228/236)	14 (6.1)	17 (7.2)
	Dementia (*n* = 116/128)	5 (4.3)	4 (3.1)
	Liver failure (*n* = 116/128)	5 (4.3)	6 (4.7)
Smoking status (*n* = 229/233)	Active	50 (21.8)	48 (20.6)
	Ex-smoker	120 (52.4)	116 (49.8)
	Non-smoker	59 (25.8)	69 (29.6)
Pack-Years (PY), median [Q1–Q3] (*n* = 136/132)		40 [30–55]	40 [25–57.5]
Alcohol consumption (*n* = 222/224)		85 (38.3)	84 (37.5)
Beer consumption (*n* = 85/84)		5 (5.9)	15 (17.9)
Wine consumption (*n* = 85/84)		51 (60.0)	40 (47.6)
Strong alcohol consumption (*n* = 85/84)		9 (10.6)	14 (16.7)
Baseline symptoms	Dysphagia (*n* = 227/234)	44 (19.4)	53 (22.7)
	Dysphonia (*n* = 227/233)	57 (25.1)	63 (27.0)
Swallowing disorders (*n* = 226/232)		9 (4.0)	10 (4.3)
	Trismus (*n* = 226/233)	2 (0.9)	7 (3.0)
Oncologic characteristics			
Cancer site (*n* = 232/230)	Oral	101 (43.5)	102 (44.4)
	Oropharyngeal	53 (22.9)	50 (21.7)
	Laryngeal	58 (25.0)	57 (24.8)
	Hypopharyngeal	20 (8.6)	21 (9.1)
T stage	Tis/T1	59 (24.8)	52 (21.9)
	T2	67 (28.1)	76 (32.1)
	T3	53 (22.3)	47 (19.8)
	T4a/T4b/Tx	59 (24.8)	62 (26.2)
N stage	N0	134 (56.3)	128 (54.0)
	N1	37 (15.6)	42 (17.7)
	N2a	16 (6.7)	17 (7.2)
	N2b	29 (12.2)	23 (9.7)
	N2c	12 (5.0)	20 (8.4)
	N3	10 (4.2)	7 (3.0)
Metastases (*n* = 226/233)	M0	180 (79.7)	184 (79.0)
	M1/Mx	46 (20.3)	49 (21.0)

Experimental arm refers to the GA-driven intervention and follow-up arm. Abbreviations: BMI, body mass index; COPD, chronic obstructive pulmonary disease; PAOD, Peripheral Arterial Obstructive Disease. ^†^ Weight loss was defined as a loss of ≥10% of bodyweight during the previous 6 months.* based on 8 items: age, appetite change, weight loss, mobility, body mass index, neuropsychological problems, self-rated health, medication.

**Table 2 cancers-14-03290-t002:** Characteristics of the geriatric assessment and GA-driven interventions in the experimental arm (*n* = 238).

Place of GA (*n* = 196)	
Geriatric day hospital	5 (2.5)
Consultation	135 (68.9)
Hospitalization	56 (28.6)
Functional status	
IADL score <8 (*n* = 193)	78 (40.4)
Mobility	
Outside walking with help (walking stick, medical walker, human help) (*n* = 196)	32 (16.3)
Falls during the last 6 months (*n* = 197)	23 (11.7)
Timed get-up-and-go test >20 s (*n* = 170)	21 (12.4)
One-leg standing test <5 s (*n* = 141)	59 (41.8)
General condition	
Asthenia (*n* = 195) *	103 (52.8)
Comorbidity	
Total CIRS-G score, median [Q1–Q3] (*n* = 175)	6 [3-9]
Nutrional status	
MNA score (*n* = 181)	
<17	20 (11.0)
[17–24]	60 (33.2)
≥24	101 (55.8)
Cognition and depression	
MMSE score <24 (*n* = 170)	31 (18.2)
Felt depressed (*n* = 196)	47 (24.0)
GA-driven interventions (n = 174)	
Unnecessary	65 (37.4)
Suggested/Prescribed by geriatrician	109 (62.6)
Social support (human or financial) (*n* = 174)	
Unnecessary	128 (73.6)
Suggested/Prescribed by geriatrician	46 (26.4)
Nursing care (*n* = 169)	
Unnecessary	148 (87.6)
Suggested/Prescribed by geriatrician	21 (12.4)
Physical therapy (*n* = 171)	
Unnecessary	133 (77.8)
Suggested/Prescribed by geriatrician	38 (22.2)
Nutritional support (dietitian visits or nutritional supplements) (*n* = 171)	
Unnecessary	87 (50.9)
Suggested/Prescribed by geriatrician	84 (49.1)
Medication review (*n* = 171)	
Unnecessary	146 (85.4)
Suggested/Prescribed by geriatrician	25 (14.6)
Memory consultation (*n* = 171)	
Unnecessary	159 (93.0)
Suggested/Prescribed by geriatrician	12 (7.0)
Psychological support (psychologist or psychiatrist) (*n* = 174)	
Unnecessary	149 (85.6)
Suggested/Prescribed by geriatrician	25 (14.4)

* Clinically assessed by the physician.

**Table 3 cancers-14-03290-t003:** Impact of GA-driven interventions and follow-up on 6-month overall survival and the functional and nutritional status of older patients with HNSCC.

	Experimental Arm *n* = 238	Control Arm *n* = 237	*p*
Intent-to-treat analysis			
Primary endpoint (death, weight loss ^a^ and decrease in ADL score ^b^) with multiple imputation	98 (41.0)	90 (38.0)	0.53 *
Death ^‡^	31 (13.0)	27 (11.4)	0.48 **
Weight loss ^a^	69 (29.0)	65 (27.4)	0.73 *
Decrease in ADL score ^b^	9 (3.8)	13 (5.5)	0.35 *
Primary endpoint (death, weight loss ^a^ and decrease in ADL score ^b^) on complete data case (*n* = 194/196)	82 (42.3)	76 (38.8)	0.48 ***
Death	31 (13.0)	27 (11.4)	0.48 **
Weight loss ^a^ (*n* = 166/172)	48 (28.9)	44 (25.6)	0.49 ***
Decrease in ADL score ^b^ (*n* = 179/183)	6 (3.4)	10 (5.5)	0.33 ***
Per-protocol analysis			
Primary endpoint (death, weight loss ^a^ and decrease in ADL score ^b^) with multiple imputations (*n* = 62/226)	19 (30.6)	86 (38.1)	0.35 *
Death ^‡^	4 (6.5)	26 (11.5)	0.27 **
Weight loss ^a^	16 (25.8)	62 (27.4)	0.79 *
Decrease in ADL score ^b^	2 (3.2)	13 (5.8)	0.55 *

Experimental arm refers to the GA-driven intervention and follow-up arm. * Logistic regression. ** Log rank test. *** Pearson’s chi-square test. ^‡^ Without imputation. ^a^ Loss of ≥10% of bodyweight during the previous 6 months. ^b^ At least 2-point decrease in the Activities of Daily Living (ADL) score during the previous 6 months.

## Data Availability

The data presented in this study are available on request from the corresponding author.

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
