# Peer review of "Effectiveness of Geriatric Assessment-Driven Interventions on Survival and Functional and Nutritional Status in Older Patients with Head and Neck Cancer: A Randomized Controlled Trial (EGeSOR)"

_cancers, 2022, doi:10.3390/cancers14133290_

Round 1

Reviewer 1 Report

I reviewed a manuscript "Effectiveness of Geriatric Assessmentdriven interventions on Survival and Functional and Nutritional Status in Older Patients with Head and Neck Cancer: a Randomized Controlled Trial (EGeSOR)" by Paillaud et al.

The aim of this study was to evaluate assess the impact of the CGA on overall survival, function, and nutritional status of elderly patients with HNSCC. This is a well conducted randomized trial of complex intervention and provides valuable information on CGA in this population.

# Major points

The authors reported the scheduled cancer treatment as follows: "The scheduled cancer treatment was surgery in 55.2% of patients in the experimental arm and 50.8% of patients in the control arm, chemotherapyradiotherapy in 15.2% and 14.4% of patients respectively, and radiotherapy alone in 12.6% and 14.4%, respectively."

I am not sure if these are the treatments the patients received in this trial.

One of the benefits of CGA in the oncology setting is to modify treatment decision according to a patient's frailty level and values and preferences.

In this study, CGA was planned to be conducted before making a final oncologic treatment decision.

How often was CGA done in this way (i.e., before treatment decision as described in the methods page 4) and how often was it done after a final treatment decision was made?

For patients who had a CGA before treatment decision, were their oncologic treatment decisions changed based on CGA findings?  and what are the changes?

I think these are key information to understand the effects of CGA on the outcomes of this study.

#Minor points

In discussion page 11, the authors wrote "The degree of implementation of intervention is rarely reported in the studies," by citing the trial results based on meeting abstracts. For reference numbers 16 and 18, papers have been published (as below) and the implementation rates of CGA interventions have been reported. I suggest that the authors read these papers and update this paragraph based on the findings from these trials.

PMID: 34591080

Geriatric assessment-driven intervention (GAIN) on chemotherapy related toxic effects in older adults with cancer: a randomized clinical trial

PMID: 34741815

Evaluation of geriatric assessment and management on the toxic effects of cancer treatment (GAP70+): a cluster-randomised study

Regarding the sentence "The principal explanation for these results may be the high proportion (74%) of patients who did not benefit from the complete intervention as planned even though all the patients in the intervention group received a GA". I could not understand where "74%" number came from. I think this sentence needs clarification.

In table 2, "asthenia" is reported, but there is no description how asthenia was assessed in this study.

In page 4, these is a sentence with incomplete reference "The ADL and weight measurements were standardized using a measurement guide and a dedicated balance, similar in every center (reference: Seca 813 Robusta, designed in Hamburg (Germany) and made in China)."

The reference 21 and 22 are the same papers.

Reviewer 2 Report

That is an interesting open-label, multicenter, randomized, controlled and parallel-group trial in which it has been analyzed the benefit of GA-driven interventions in older patients with head and neck cancer. 

There are many aspects of concern: it was necessary to increase the inclusion of patients because of the low inclusion. And most important: they have been included patients with many kind of treatments: the scheduled cancer treatment was surgey in 55.2% of patients in the experimental arm and 50.8% of patients in the control arm; and chemotherapy-radiotherapyy in 15.2% and 14.4% of patients respectively, and radiotherapy alone in 12.6% and 14.4% respectively. In my opinion the primary outcome cannot be achieved if there are three different kinds of treatment in the population, as GA-driven interventions are probably different for each kind of treatment and the expected outcomes are also different. The statistical analysis should be made separately for any kind of treatment.

Reviewer 3 Report

It is my pleasure to review this manuscript.

This is a randomized controlled trial to evaluate the benefits of GA driven interventions and follow‐up on 6‐month overall survival and functional and nutritional status in older patients with HNSCC and found that in older patients with HNC, GA‐driven interventions and follow‐up failed to improve 6‐month overall survival, functional and nutritional status. I have following major concerns and comments:

1.      More detailed on GA assessment is needed, is it a comprehensive GA assessment as described in introduction or a regular GA. More details on each measurement of GA should also be provided or referenced. For example, rationale of using MNA instead of other tools for malnutrition. What is the tool for frailty? There should be section for both GA assessment and intervention.

2.      Line 198-200, “unplanned futility …” is very confusing, and how?

3.      The biggest issue includes the change of eligibility criteria, failed to recruit the proposed sample size for analysis (might not have enough power to detect the difference, the real power should be provided). In this sense, if the power is big issue, all the results are not convincing.

4.      There are several other issues: the statistical analysis is not strong, for example, for mortality data, it is usually conducted in a cox regression analysis to take consideration of time. Missing data imputation on what variables is not clear and its criteria and strategy is not provided.

5.      What does G-8 stand for? This not clear in the manuscript.

6.      For nutrition status, why did the author use weight loss, instead of MNA as the outcome indicator?

7.      A potential selection bias due to withdraw/loss to follow up might need to be assessed in a supplemental table.

Round 2

Reviewer 1 Report

The authors updated their manuscript according to the reviewers’ comments.

On page 2 of updated cover letter, the authors report "We agree with the remark but we did not collect the information on the initial ENT physician proposal before CGA." I think this is an important limitation of this study and I would like the authors to mention this in the discussion section.

Reviewer 2 Report

Authors have greatly improved the manuscript

Author Response

We thank the reviewer for the comment.

Reviewer 3 Report

The authors have addressed all my comments and concerns well. 

Author Response

We thank the reviewer for the comment.